# Effects of additional load at different heights on gait initiation: A statistical parametric mapping of center of pressure and center of mass behavior

**Marcus Fraga Vieira**[1], **Fábio Barbosa Rodrigues**[1,2], **Alfredo de Oliveira Assis**[1], **Eduardo de Mendonça Mesquita**[1], **Thiago Santana Lemes**[1], **Guilherme Augusto Gomes De Villa**[1], **Rafael Reimann Baptista**[3]\*, **Adriano de Oliveira Andrade**[4], **Paula Hentschel Lobo da Costa**[5]

**1** Bioengineering and Biomechanics Laboratory, Federal University of Goiás, Goiânia, Goiás, Brazil, **2** State University of Goiás – Unidade Trindade-GO, Brazil, **3** Pontifical Catholic University Rio Grande do Sul, Porto Alegre, Brazil, **4** Centre for Innovation and Technology Assessment in Health (NIATS), Federal University of Uberlândia, Uberlândia, Brazil, **5** Physical Education Department, Federal University of São Carlos, São Carlos, Brazil

\* baprafael@gmail.com

## Abstract

The purpose of this study was to investigate the effects of different vertical positions of an asymmetrical load on the anticipatory postural adjustments phase of gait initiation. Sixty-eight college students (32 males, 36 females; age: 23.65 ± 3.21 years old; weight: 69.98 ± 8.15 kg; height: 1.74 ± 0.08 m) were enrolled in the study. Ground reaction forces and moments were collected using two force platforms. The participants completed three trials under each of the following random conditions: no-load (NL), waist uniformly distributed load (WUD), shoulder uniformly distributed load (SUD), waist stance foot load (WST), shoulder stance foot load (SST), waist swing foot load (WSW), and shoulder swing foot load (SSW). The paired Hotelling's T-square test was used to compare the experimental conditions. The center of pressure (COP) time series were significantly different for the SUD vs. NL, SST vs. NL, WST vs. NL, and WSW vs. NL comparisons. Significant differences in COP time series were observed for all comparisons between waist vs. shoulder conditions. Overall, these differences were greater when the load was positioned at the shoulders. For the center of mass (COM) time series, significant differences were found for the WUD vs. NL and WSW vs. NL conditions. However, no differences were observed with the load positioned at the shoulders. In conclusion, only asymmetrical loading at the waist produced significant differences, and the higher the extra load, the greater the effects on COP behavior. By contrast, only minor changes were observed in COM behavior, suggesting that the changes in COP (the controller) behavior are adjustments to maintain the COM (controlled object) unaltered.

**Data Availability Statement:** The data underlying the results presented in the study are available from: http://doi.org/10.5281/zenodo.4250894.

**Funding:** The study was partially supported by the governmental agencies CAPES (Finance Code 001), and CNPq.

**Competing interests:** The authors have declared that no competing interests exist.

## 1. Introduction

Gait initiation (GI) is the functional task of transitioning from a standing position to a new cyclic walking. It is delimited by three phases: the initial swing foot loading phase resulting from a backward and lateral shift in the center of pressure (COP) toward the swing foot; the swing foot unloading phase, when the COP moves toward the support foot; and the support-foot unloading phase, when the COP moves forward under the support foot. The first phase is the anticipatory postural adjustments (APA) phase, which aims to set the initial conditions to stabilize the body while moving the center of mass (COM) forward and toward the support foot to execute the first step of a new gait cycle [1].

Even before any detectable foot movement, APA mechanism acts as a first stabilizing strategy to prevent mediolateral imbalances by shifting the center of pressure towards the swing foot, which in turn shifts the center of mass to the stance foot. In the standing position COP and COM are close to each other. Before GI begins, ankle muscles of both lower limbs are active to maintain COM in a stable position between the feet. Now the task is to move the body outside the base of support and, at the same time, prevent falling [2]. GI begins with the posterior and lateral displacement of the COP toward the foot that will move first, namely the swing foot. This increases the anterior-posterior (AP) and medial-lateral (ML) ground reaction forces that, in turn, cause the displacement of the COM in the opposite direction, forward and toward the support foot [3–5]. The result is that COP and COM begin to dissociate. During the APA phase, the COP backward movement and toward the swing limb is considered an efficient strategy, because it generates the forward impulse needed to begin walking without requiring the COM to move out of the base of support and the lateral impulse needed to shift the center of mass toward the support foot [6].

In the sagittal plane, the AP COP displacement during APA phase is mediated by the stereotyped activities of ankle muscles: the inhibition of the soleus and gastrocnemius that are followed by activation of the tibialis anterior in both legs [7]. The anticipatory soleus inhibition and tibialis anterior activation are not observed in all young healthy adults [8] and this functional variability of APA behavior is probably influenced by initial trunk posture (backward or forward inclination), speed of the first step and initial tonic muscle soleus activity [8].

In the frontal plane, the ML COP displacement during GI are controlled by coordinated action of swing leg hip abductors [2]. While COP moves backward by ankle muscles synergism, swing limb hip abductors move the COP to the swing limb [2] during the APA phase. Further studies have reported a slight knee and hip flexion of the support limb during APA that acts to unload the support limb complementing the action of the swing limb hip abductors [9]. In addition, it was observed that ankle dorsiflexors contribute to ML COP displacement during APA: coordinated activation of hip abductors and tibialis anterior during APA has a role in the ML COP displacement towards the swing foot [9].

Given the functional importance of mastering the transition from standing to a new gait cycle, some researchers [10–13] have explored the mechanism of APA adaptation to asymmetrical overloading and how this condition affects postural stability. Frontal plane instabilities during GI can be increased in a condition of asymmetrical bodyweight distribution or when loads are asymmetrically added to the body. The first condition can be found in pathological patients, and the second condition can be found in ecological situations such as carrying a tool belt, or a backpack on one side.

Studies have shown that when manipulating the frontal plane weight distribution by shifting the body weight to the swing foot side or by overloading the same side with an extra load attached to a waist belt prior to gait initiation, APA duration became longer and forward progression velocity faster compared to a situation with symmetrical weight distribution

[10,11,13]. By contrast, when weight distribution was shifted to the support foot side, the results were the opposite. Thus, the amplitude of the medial-lateral (ML) COP shift to the swing foot side during APA changes according to the weight distribution on the frontal plane.

However, there were conflicting results in the aforementioned studies regarding adaptations of the APA mechanism in the anterior-posterior (AP) direction following ML weight shift prior to gait initiation. While some authors have found that APA parameters in the AP direction changed following bodyweight distribution between the lower limbs [10,13,14], similar progression velocities used across conditions appeared to prevent the same parameters from changing in order to maintain ML stability during gait initiation [11].

More recently, the hypothesis of coupling between COP shifts in the ML and AP directions during gait initiation was explored after experimentally manipulating the initial COP coordinates by also shifting the body weight toward the toes and heels on the sagittal plane [15]. These experiments evidenced only a weak coupling and, more surprising, a persistent ML COP shift to the swing foot even when the whole-body weight was placed over the supporting foot before gait initiation started. Furthermore, whether COP shifts during gait initiation are mechanically coupled may be further explored by also manipulating the vertical position of an asymmetrical load in order to change body weight torques in the sagittal and frontal planes, leading to eventually more pronounced changes in the AP and/or ML COP shifts prior to gait initiation than just shifting the body weight forward or backward.

Therefore, the purpose of this study is to investigate the effects of different vertical positions of an asymmetrical load on the APA phase of gait initiation. We hypothesized that the asymmetrical weight distribution applied to different heights prior to gait initiation will elicit scaling in the COP behavior in both the AP and ML directions. Conditions with a load positioned at the stance foot and swing foot sides in both the waist and shoulder were compared with a control condition with no load. To capture the features of the entire COP and COM time series during APA (rather than discrete variables), we conducted a vector analysis using statistical parametric mapping (SPM) methods (Pataky et al., 2014).

## 2. Methods

### 2.1 Subjects and ethics statement

This study enrolled 68 college students (32 males, 36 females; age: 23.65 ± 3.21 years; weight: 69.98 ± 8.15 kg; height: 1.74 ± 0.08 m), not engaged in any mode of physical training. All students were healthy, with no history of any functional impairment, neurological or orthopedic condition, or any musculoskeletal injury or pain at the time of data collection. This study was approved by the Universidade Federal de Goiás Ethics Committee for Human Research. The form of consent obtained from the participants was written.

### 2.2 Data collection

Ground reaction forces (GRFs) and moments were collected using two force platforms (OR6-7 model, AMTI) placed at the beginning of a 4-m level walkway. Additionally, to describe the initial COP position when in a standing position in relation to a foot coordinate system, reflective markers were attached to both lateral malleoli. Both kinetic and kinematic data were captured by a motion capture system (Vicon Motion Systems Ltd, Oxford, UK) comprising 10 infrared cameras and the two force platforms, operating synchronously at 100 Hz.

The participants were instructed to assume a comfortable and natural standing position, barefoot, upper limbs alongside the trunk, one foot on each force platform. The participants were tested barefoot to avoid any influence of footwear and to make our results comparable to that of previous studies [11,16–18]. They were asked to stand as still as possible. After a verbal

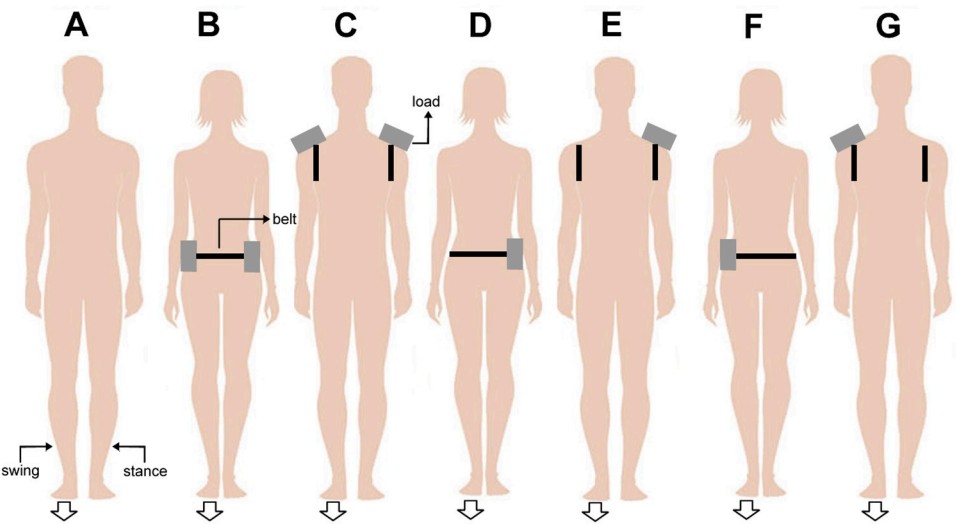

**Fig 1. Schematic representation of experimental conditions: (A) no load (NL), (B) waist uniformly distributed load (WUD), (C) shoulder uniformly distributed load (SUD), (D) waist stance foot load (WST), (E) shoulder stance foot load (SST), (F) waist swing foot load (WSW), and (G) shoulder swing foot load (SSW).** The arrow indicates the foot used to start the movement (swing foot).

command, the participants started to walk with their preferred foot at a self-selected speed to the end of the walkway, performing two complete gait cycles. The length of the first step was not controlled in order not to impose restrictions on any change in the task execution strategy due to the different conditions. Next, they were asked to reposition themselves after each trial following the same instructions, taking the midline between the force platforms as a reference, and starting with the same foot. They executed as many familiarization trials as needed. The data acquisition started 2 s prior to the verbal command. The participants rested for a period of 30 s between each trial.

The participants executed three trials under each of the following conditions: no load (NL), waist uniformly distributed load (WUD), shoulder uniformly distributed load (SUD), waist stance foot load (WST), shoulder stance foot load (SST), waist swing foot load (WSW), and shoulder swing foot load (SSW) (Fig 1). The experimental conditions were randomly assigned to each participant. When present, the additional load was set to 10% of body mass [11,19]. The load consisted of a belt firmly positioned at the waist (close to the COM's position) or at the shoulder. Weights were attached to the belt to carefully reach the desired load.

Most participants started to walk with the right foot. When the participant started to walk with the left foot, the ML COP sign was reversed for an appropriate comparison [20]. The beginning of the task was identified as the first ML COP deviation toward the swing foot greater than 3 standard deviations of the mean COP position at the 1.5 s prior to the verbal command [21]. The end of the APA was defined as the greatest ML COP shift toward the swing foot.

## 2.3 Data analysis

The motion capture system software (Vicon Nexus, version 2.11, Vicon Motion Systems Ltd, Oxford, UK) provided the COP, ground reaction forces (GRF), and kinematic time series. The raw force platform and kinematic data were filtered using a fourth-order, zero-lag, low-pass Butterworth filter with a cutoff frequency of 12.5 Hz [22]. The COP, GRF, and kinematics time series were exported in txt file (a full description of how to calculate the COP can be found

elsewhere [16,23]). Next, the COP time series were normalized such that at the beginning of the task, the AP and ML COP components were described in relation to the mid-point between the lateral malleoli (i.e., in relation to a foot coordinate system). The projection of COM on the force platform was estimated using a double integration of the previously exported GRFs by the trapezoidal method [22] using a custom-written Matlab code (© 1994–2021 The MathWorks, Inc.).

The APA phase was identified according to Vieira et al. [6,16,24]. A vector analysis of the resulting COP and the estimated COM during the APA phase was conducted using the SPM method [25]. This statistical approach captures features of the entire time series, rather than discrete variables, and it may provide additional information for gait initiation analyses. In contrast, discrete variables fail to capture sufficient portions of the data and covariance among vector components [25]. SPM analysis uses random field theory to identify field regions that co-vary with the experimental protocol [26,27].

Each AP and ML component of each APA time series (COP and COM) was interpolated with *pchirp* to contain 61 points (corresponding to approximately 0–30% of the entire task). The average obtained over the three trials was used in the analysis. Next, for each APA time series, the AP and ML components were organized in an array with two corresponding matrices, containing 68 rows (one for each subject) and 61 columns. One array was constructed for each experimental condition.

The paired Hotelling's T-square test (the SPM vector field analog to the paired t-test) was used for comparing the experimental conditions. Paired t-tests were conducted as a *post-hoc* test, with a Sidák correction (Eq 1) producing $p_{critical}$ = 0.0253 for $N$ = 2 and $\alpha$ = 0.05:

$$p_{critical} = 1 - (1 - \alpha)^{1/N} \tag{1}$$

The output of SPM provided the T-square and t values for each sample of the COP and COM time series, and the threshold corresponding to $\alpha$ was set at 0.05. The values of T square and t above the threshold (shadow portions in Figs 4–9) indicated significant differences in the corresponding portion of the APA time series.

The Matlab codes provided by www.spm1d.org were used to conduct the SPM analysis. They were inserted into a custom-written Matlab program to process the data and to plot the graphs.

## 3. Results

The average COP APA phase displacement is shown in Fig 2, where the initial COP displacement is toward the swing foot (right foot in this case). The greatest positive differences were observed for the WSW condition (Fig 2A) and the greatest negative differences for the SST condition (Fig 2C). For both, the greatest difference occurred at the end of the APA phase in the ML direction.

The average COM APA phase displacement is shown in Fig 3, where the initial COM displacement was toward the support foot. The greatest differences were observed in the AP direction with the additional weight at the waist for both the WUD and WSW conditions.

### 3.1 Effect of additional symmetrical and asymmetrical load at the waist

In the SPM analysis of the COP time series with the weight positioned at the waist (Fig 4), differences were observed only in the WST and WSW conditions. No differences were observed during the APA for the WUD and NL conditions. The weight positioned at the stance foot side (WST) produced a smaller ML COP displacement at the beginning of the APA phase

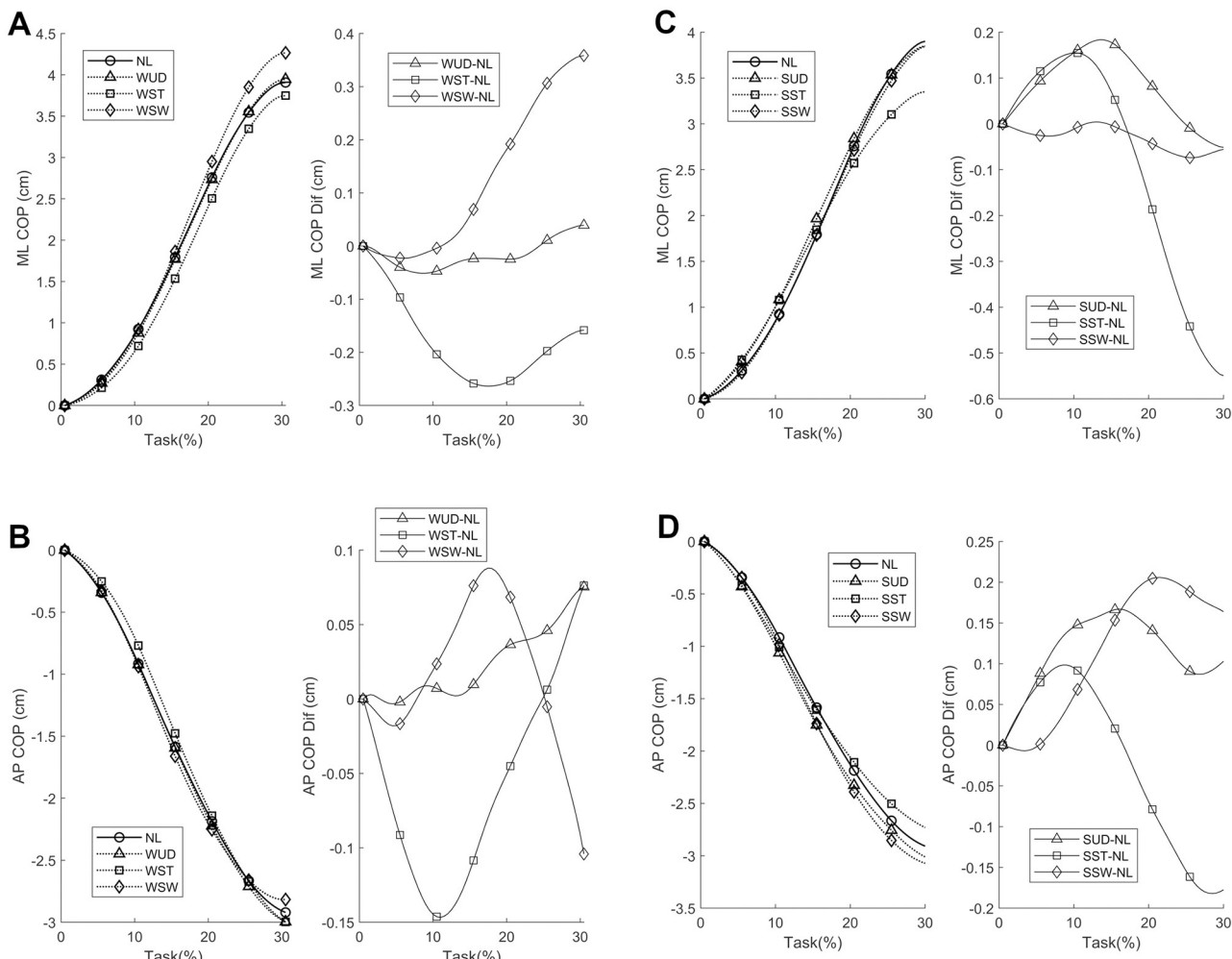

**Fig 2. Average medial-lateral (ML) and anterior-posterior (AP) center of pressure (COP) anticipatory postural adjustments (APA) phase components, with additional weight positioned at the waist (A and B, respectively) and shoulder (C and D, respectively) and their corresponding differences compared to the no-load (NL) condition.** WUD: weight uniformly distributed at the waist; WST: weight positioned at the waist on the stance foot side; WSW: weight positioned at the waist on the swing foot side, SUD: weight uniformly distributed at the shoulder; SST: weight positioned at the shoulder on the stance foot side; SSW: weight positioned at the shoulder on the swing foot side.

compared to the NL condition, indicating a lower ML COP velocity. This situation was also observed in the AP direction, although to a lesser extent. The weight positioned at the swing foot side (WSW) produced a greater ML COP displacement at the end of the APA phase when compared to NL condition, indicating a greater APA phase. No differences were observed in the AP direction.

In the SPM analysis of the COM time series with the weight positioned at the waist (Fig 5), differences were observed between the WUD and WSW conditions. The WUD condition presented a greater COM displacement in the AP direction at the end of the APA phase when compared to the NL condition. In the WSW condition, COM displacement was greater in both the ML and AP directions at the end of the APA phase compared to the NL condition, indicating a greater COM repositioning with the weight positioned on the swing foot side.

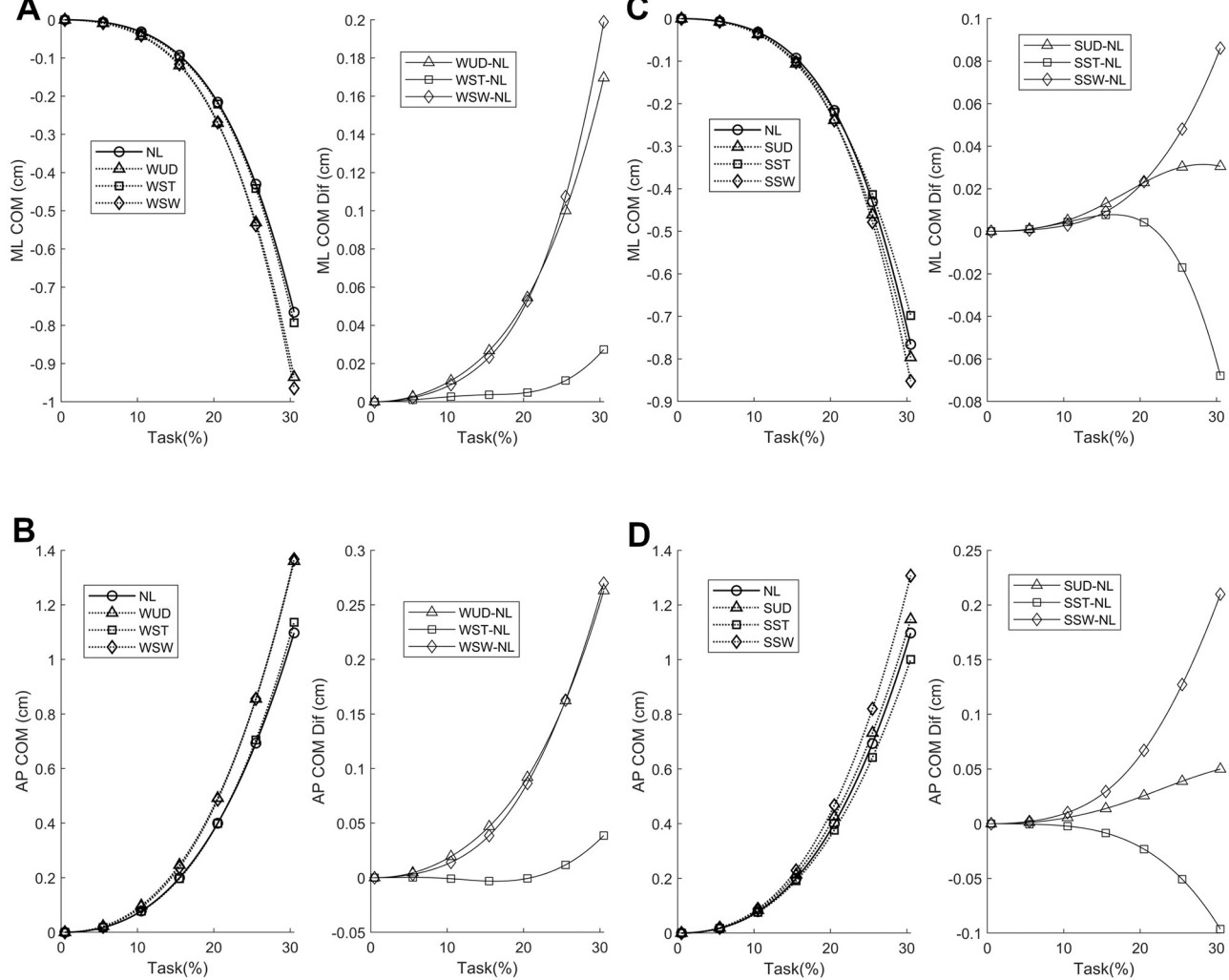

**Fig 3. Average medial-lateral (ML) and anterior-posterior (AP) center of mass (COM) anticipatory postural adjustments (APA) phase components, with additional weight positioned at the waist (A and B, respectively) and shoulder (C and D, respectively) and their corresponding differences compared to the no-load (NL) condition.** WUD: weight uniformly distributed at the waist; WST: weight positioned at the waist on the stance foot side; WSW: weight positioned at the waist on the swing foot side, SUD: weight uniformly distributed at the shoulder; SST: weight positioned at the shoulder on the stance foot side; SSW: weight positioned at the shoulder on the swing foot side.

### 3.2 Effect of additional symmetrical and asymmetrical load at the shoulder

In the SPM analysis of the COP time series with the weight positioned at the shoulder (Fig 6), differences were observed in the SUD and SST conditions. The greatest differences were observed in the ML direction. The SUD condition produced greater ML COP displacement at the beginning of the APA phase compared to the NL condition, indicating a greater ML COP velocity. A similar result was observed in the AP direction, although to a lesser extent. The SST condition produced similar results; a greater ML COP displacement was found at the beginning of the APA phase compared to the NL condition, indicating a greater ML COP velocity. A similar result was found for the AP direction, although to a lesser extent. No differences were found between the SSW and NL conditions.

In the SPM analysis of the COM time series with the weight positioned at the shoulder (Fig 7), no differences were observed in any of the comparisons.

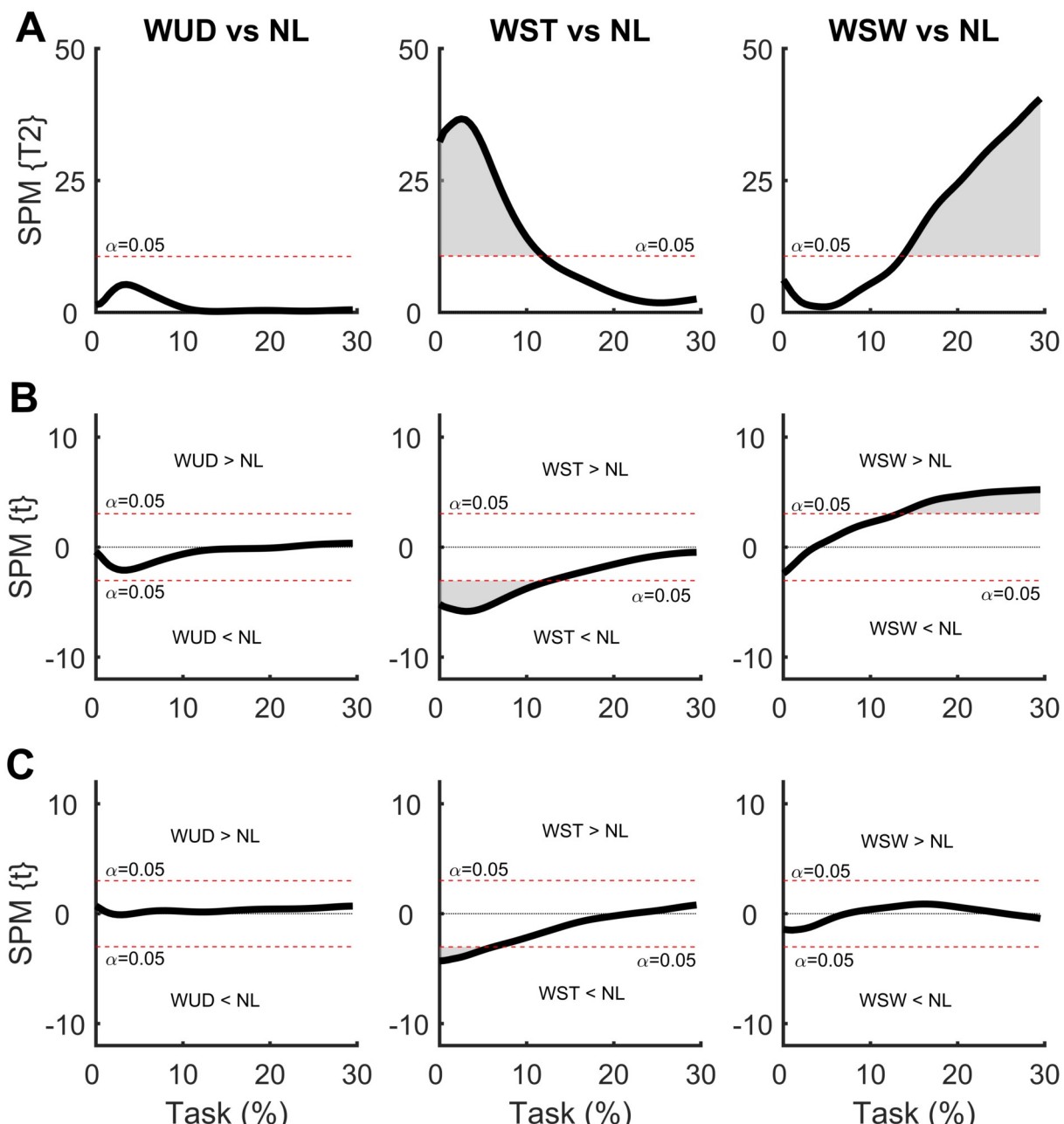

**Fig 4.** **(A)** Hotelling's paired T-square test on center of pressure (COP) displacement vector with the weight positioned at the waist. Post hoc scalar t-tests on the **(B)** medial-lateral (ML) COP and **(C)** anterior-posterior (AP) COP components. NL: no-load; WUD: weight uniformly distributed at the waist; WST: weight positioned at the waist on the stance foot side; WSW: weight positioned at the waist on the swing foot side.

### 3.3 Effect of additional symmetrical and asymmetrical load at different heights

In the SPM analysis comparing COP time series with the weight positioned at the waist and at the shoulder (Fig 8), differences were observed between all comparisons. For the weight uniformly distributed, the SUD condition presented a greater ML COP displacement at the

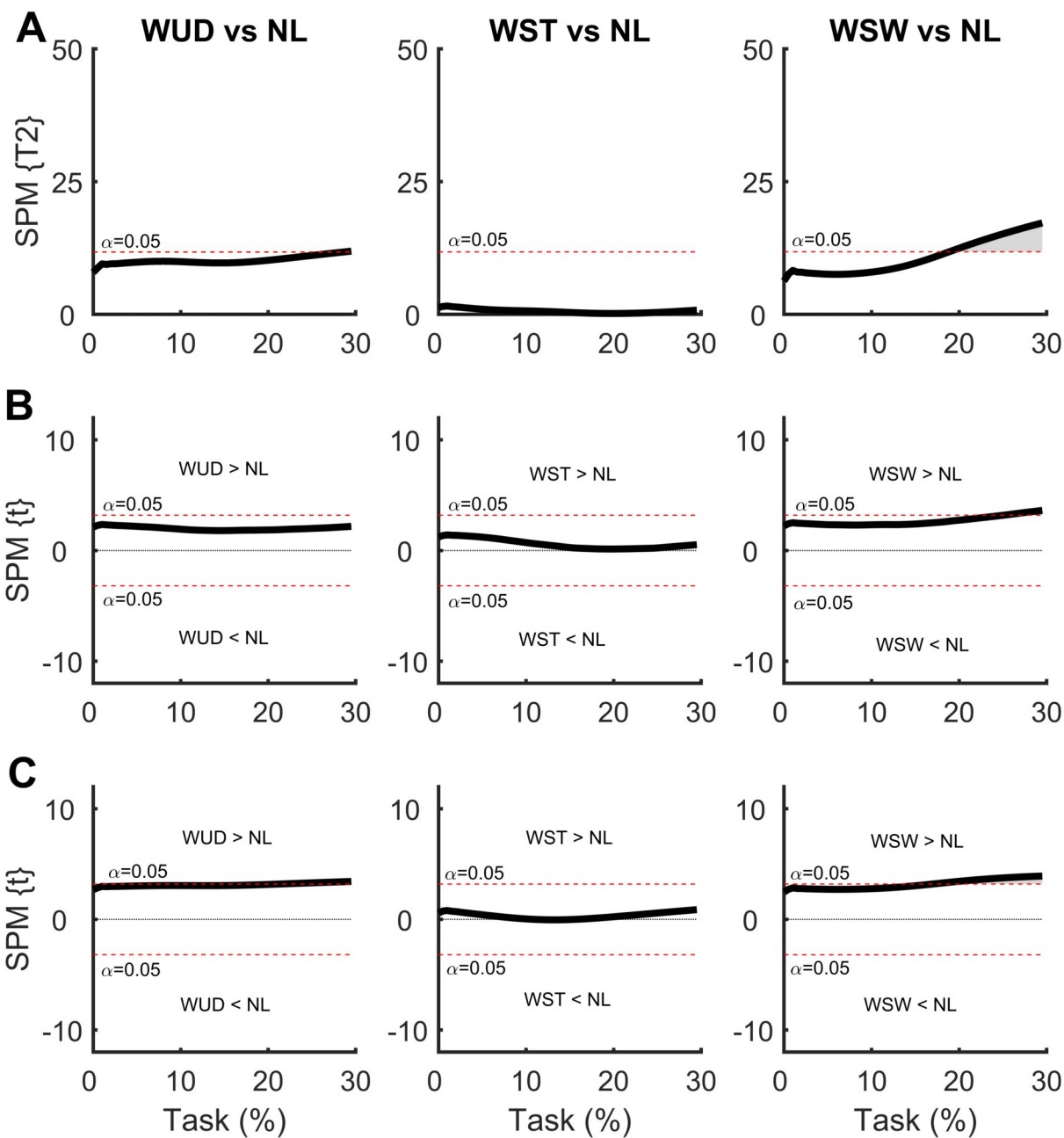

**Fig 5.** **(A)** Hotelling's paired T-square test on center of mass (COM) displacement vector with the weight positioned at the waist. Post hoc scalar t-tests on the **(B)** medial-lateral (ML) COP and **(C)** anterior-posterior (AP) COP components. NL: no-load; WUD: weight uniformly distributed at the waist; WST: weight positioned at the waist on the stance foot side; WSW: weight positioned at the waist on the swing foot side.

beginning of the APA phase compared to the WUD condition, indicating greater ML COP velocity with the weight positioned at the shoulder. A similar result was found in the AP direction, but to a much lesser extent.

Asymmetrical load on the stance foot side resulted in greater ML COP displacement at the beginning of the APA phase in the SST condition, which was indicative of a greater ML COP velocity, with the same result found in the AP direction. Contrasting results were found with

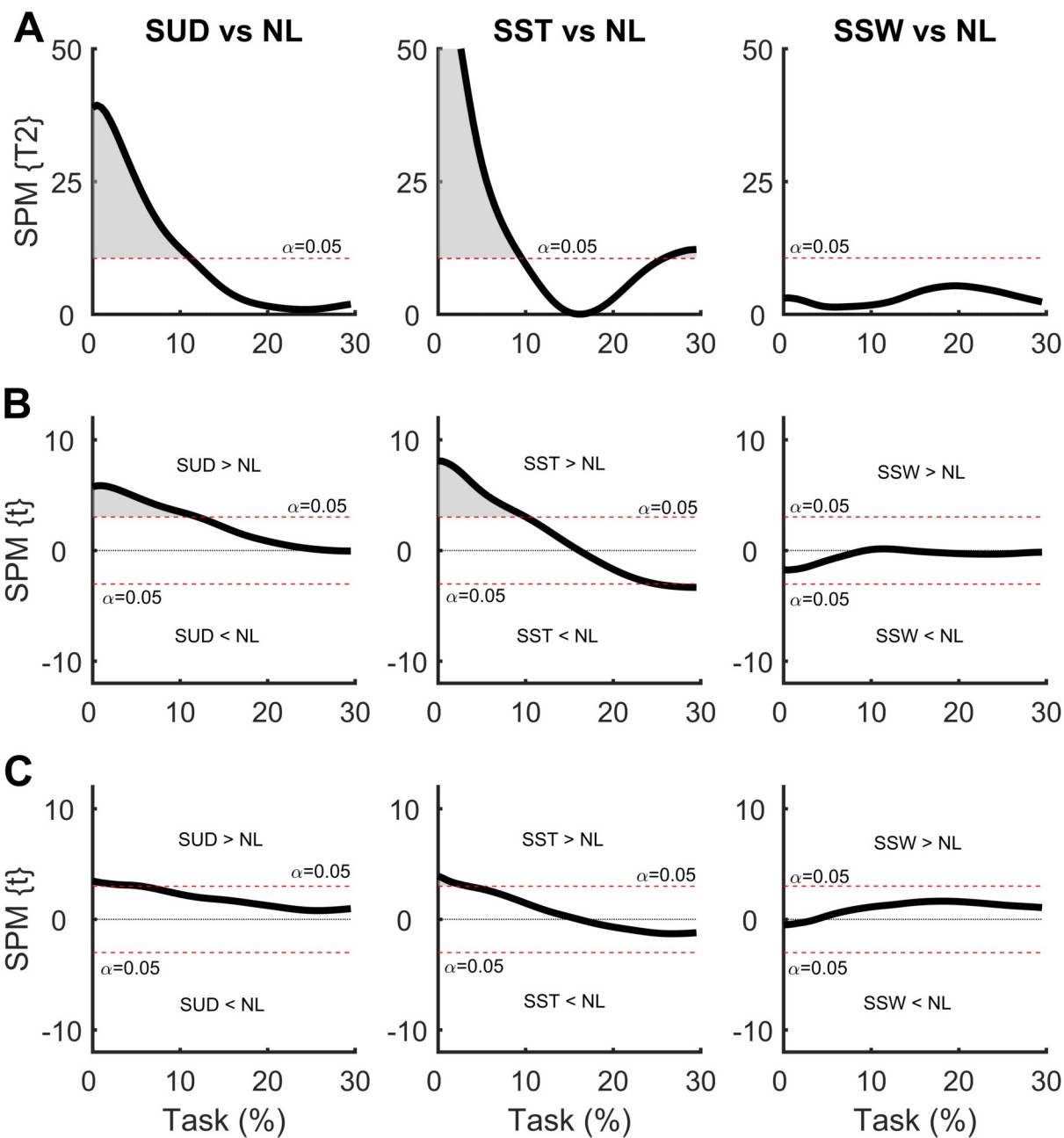

**Fig 6.** **(A)** Hotelling's paired T-square test on center of pressure (COP) displacement vector with the weight positioned at the shoulder. Post hoc scalar t-tests on the **(B)** medial-lateral (ML) COP and **(C)** anterior-posterior (AP) COP components. NL: no-load; SUD: weight uniformly distributed at the shoulder; SST: weight positioned at the shoulder on the stance foot side; SSW: weight positioned at the shoulder on the swing foot side.

the weight positioned on the swing foot side; the WSW condition presented greater ML COP displacement at the end of the APA phase, indicating a greater APA phase. No differences were observed in the AP direction.

In the SPM analysis comparing the COM time series with the weight positioned at the waist and the shoulder (Fig 9), no differences were observed for any comparisons.

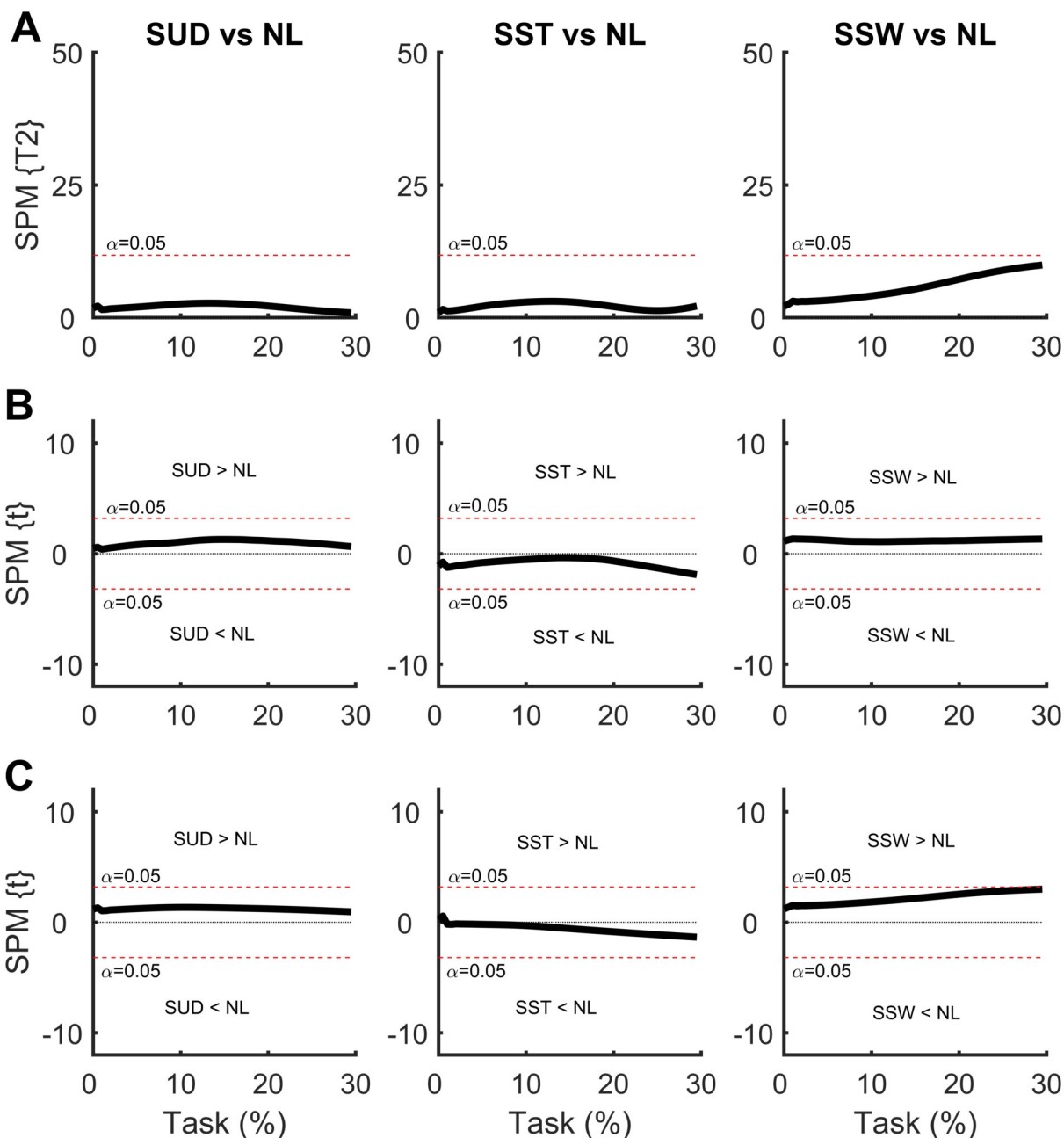

**Fig 7. (A)** Hotelling's paired T-square test on center of mass (COM) displacement vector with the weight positioned at the shoulder. Post hoc scalar t-tests on the **(B)** medial-lateral (ML) COP and **(C)** anterior-posterior (AP) COP components. NL: no-load; SUD: weight uniformly distributed at the shoulder; SST: weight positioned at the shoulder on the stance foot side; SSW: weight positioned at the shoulder on the swing foot side.

## 4. Discussion

This study compared the effects of different vertical positions of an asymmetrical load on the APA phase of gait initiation through the SPM analysis of the COP and COM. SPM analysis captures features of the entire time series, rather than a few discrete variables, so it may provide additional information missed by discrete variables. Our hypothesis was that an asymmetrical

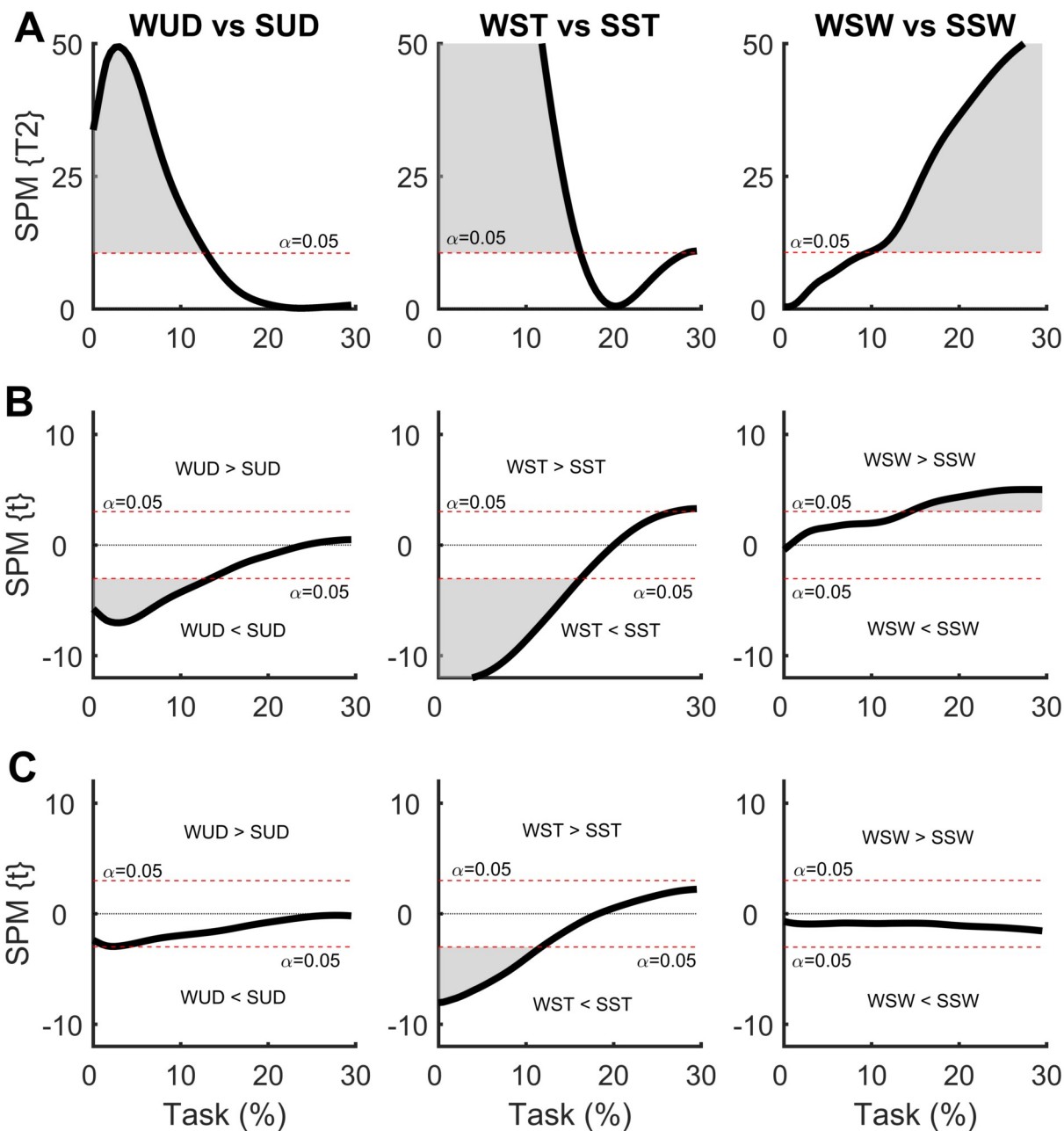

**Fig 8.** **(A)** Hotelling's paired T-square test on the center of pressure (COP) displacement vector comparing the weight positioned at the waist with the weight positioned at the shoulder. Post hoc scalar t-tests on **(B)** medial-lateral (ML) COP and **(C)** anterior-posterior (AP) COP components. WUD and SUD: weight uniformly distributed at the waist and shoulder, respectively; WST and SST: weight positioned at the waist and shoulder, respectively, on the stance foot side; WSW and SSW: weight positioned at the waist and shoulder, respectively, on the swing foot side.

weight distribution applied to different body heights before gait initiation would produce modifications in the COP and COM behavior in both the AP and ML directions.

Our main findings were that when the load is symmetrically positioned, only the highest vertical position (the shoulders) affected the COP patterns. This result suggests that when the weight is well-distributed in a lower position of the body, there are fewer perturbations to the

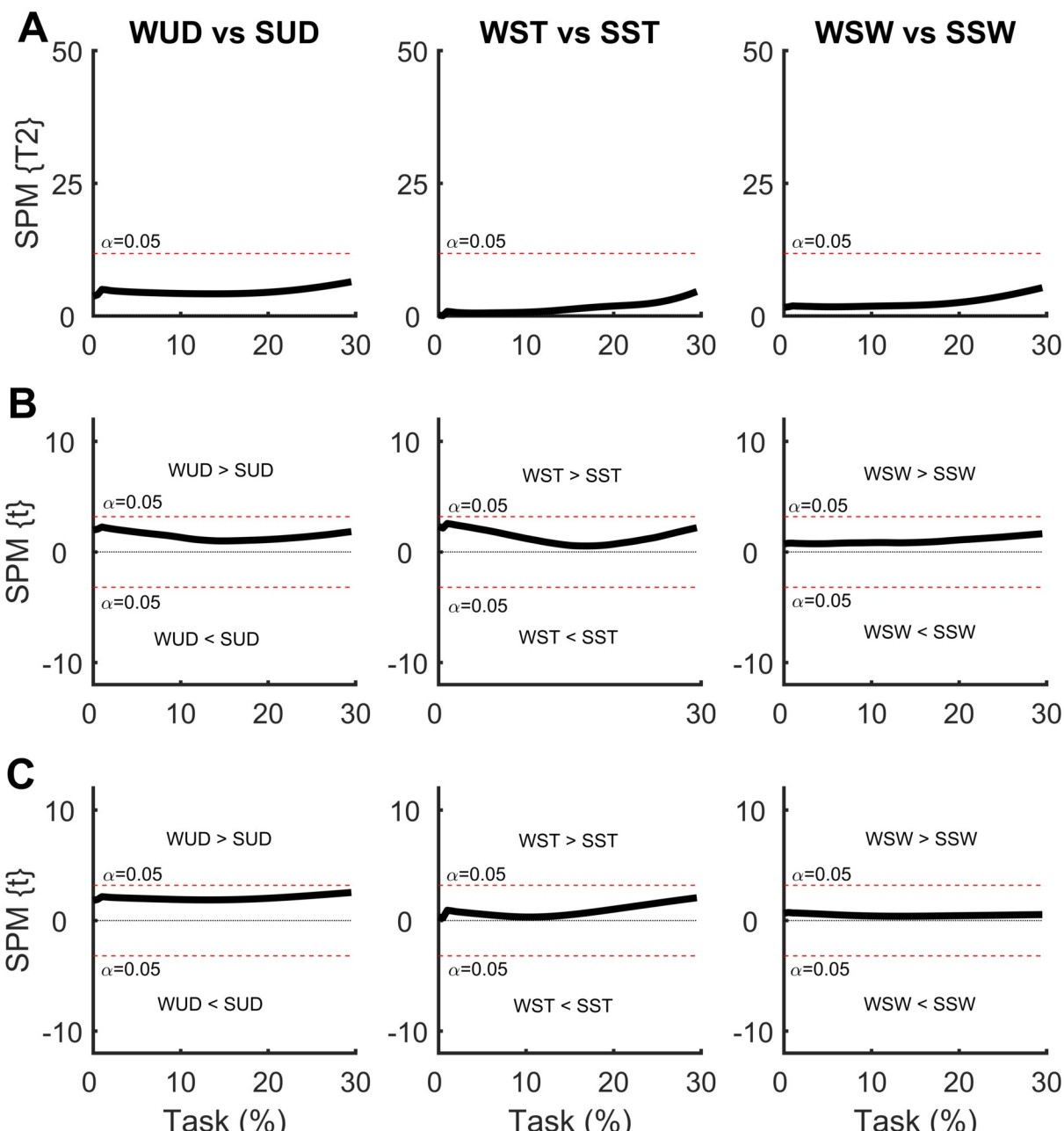

**Fig 9.** **(A)** Hotelling's paired T-square test on the center of mass (COM) displacement vector comparing the weight positioned at the waist with the weight positioned at the shoulder. Post hoc scalar t-tests on **(B)** medial-lateral (ML) COP and **(C)** anterior-posterior (AP) COP components. WUD and SUD: weight uniformly distributed at the waist and shoulder, respectively; WST and SST: weight positioned at the waist and shoulder, respectively, on the stance foot side; WSW and SSW: weight positioned at the waist and shoulder, respectively, on the swing foot side.

APA since the COM position is not altered compared to the NL condition. In addition, minor changes were observed in the COM patterns. This suggests that the changes in COP (controller) patterns are adjustments to maintain the COM (controlled) unaltered [3].

When the weight is at the waist on the side of the stance foot, the COM is positioned nearer to it so that a smaller COP ML APA component is required for positioning the COM on the

stance foot side for a safe first step. In contrast, when the weight is on the swing foot side, the COM is positioned farther from the stance foot, thus a greater COP ML APA component is required for positioning the COM on the stance foot side for a safe first step. The AP COP displacement was smaller in the WST compared to the NL condition, with a smaller initial AP COP speed, suggesting that the weight positioned at the waist on the stance foot side requires less initial AP COP displacement and, hence, less AP propulsion for a proper forward progression of COM (see Fig 7).

Following these observations, no changes were observed in COM displacement when comparing the WST condition to the NL condition. However, ML and AP COM displacement were greater in the WSW condition compared to the NL condition. When the weight is positioned on the swing foot side, the COM is nearer to the swing foot, and thus a greater effort would be expected to displace the COM toward the stance foot for a safe first step, as was observed. By contrast, a greater AP COM displacement in the WSW condition would be expected to result by a greater forward inclination of the body for a faster repositioning of the swing foot during the first step in order to maintain balance, considering that a corresponding AP COP change in this condition was not observed (Fig 4).

When the weight was positioned at the shoulder, changes in the COP pattern were observed in the symmetrical condition, suggesting the importance of the height of the additional load. The ML COP displacement was greater at the beginning of the APA phase when comparing the SUD vs. NL conditions. This suggests the necessity of a greater ML COP displacement to overcome the body's inertia, particularly in the ML direction, when the COM is in a higher position. In a study of high school students using a backpack (i.e., with an extra load positioned at the shoulders), Vieira et al. [28] found that the mediolateral COP displacement was larger during the first phase of the gait initiation using a bilateral backpack (i.e., with the weight uniformly ML distributed in the shoulders). This agrees with our study since the altered behavior of the GI reflects the necessity to compensate for changes in the initial COM position.

Vieira et al. [28] also found that having the weight at the shoulders on the stance foot side produces the same effect, although a greater trunk repositioning is necessary to maintain the COM at an initial stable position. This was also the case in the present study; when the weight was positioned at the shoulder on the stance foot side, we found a greater ML COP at the beginning of the APA phase to overcome the body inertia. This conclusion is supported when comparing the WST vs. SST conditions: the COM presented the same displacement during the APA phase, but a greater COP displacement was observed in the SST condition. Taken together, the findings discussed above suggest a biomechanical strategy in order to place the COM vertical projection closest to the support foot in an attempt to balance an asymmetric distribution of an extra load during the APA phase.

However, the COP behavior did not change when comparing the SSW and NL conditions when the weight was positioned at the shoulder. This result suggests that when the weight is applied to a higher position on the body on the swing foot side, no changes in COP displacement are required due to repositioning of the trunk (which is laterally inclined toward the stance foot), and other mechanisms related to the body's inverted pendulum behavior are present. The fact that the SUD condition presented a greater ML COP displacement at the beginning of the APA phase compared to the SSW condition reinforces this conclusion. Together, these findings suggest that the human body behaves like a double inverted pendulum in the ML direction, which is supported by the fact that changes were observed in both the ML and AP COM displacement with the weight positioned at the shoulder (Fig 8).

Caderby et al. [11] demonstrated that the peak of the mediolateral COP displacement toward the swing foot side during APA seems to be scaled as a function of the initial body weight distribution over the lower limbs in order to maintain the ideal conditions for stability

during GI. However, because these authors did not use SPM to analyze the effects of even and uneven extra loads applied to the waist, their findings may not be completely comparable to ours. Nevertheless, they found that the mediolateral location of the COM during APA was influenced by the load distribution, suggesting alterations in the gait initiation and in the initial distribution of the body weight between the swing and support foot sides according to the symmetry of the carried load, which is in agreement with the findings of our study.

The present study had some limitations. We only used kinetic data to describe the COP and COM displacement. Thus, we cannot make further assumptions about additional mechanisms that might be involved in force/impulse generation. Consequently, future studies should consider the simultaneous analysis of full body kinematic and electromyographic data to investigate the influence of different vertical positions of asymmetrical/symmetrical loads on the APA phase of gait initiation.

Lastly, the present study produced information regarding APA in several situations with a potential adverse effect on gait initiation in a very wide population that usually carries bags, backpacks, or tool belts for daily activities, creating an asymmetrical/symmetrical overload at different heights in the standing position. The results found here can be used for manipulating/inducing gait initiation strategies in a clinical context, especially regarding cases of pathological patients unable to maintain a symmetrical body weight distribution [29–31]. Additionally, since we have demonstrated here that asymmetrical overload can influence mediolateral postural stability during GI, and since mediolateral instability is believed to cause sideways falls, which can produce severe injuries in older adults [32], future studies should investigate if elderly people can adjust their APA to uneven extra load distribution on the body to preserve ML stability during GI. Such studies could provide the identification of APA alterations under challenging postural adaptations in older adults, could produce a better understanding of the mechanisms responsible for recurrent falls in older adults during gait initiation [33], and could even be used in rehabilitation protocols.

## Author Contributions

**Conceptualization:** Marcus Fraga Vieira, Fábio Barbosa Rodrigues, Alfredo de Oliveira Assis, Paula Hentschel Lobo da Costa.

**Data curation:** Marcus Fraga Vieira, Fábio Barbosa Rodrigues, Alfredo de Oliveira Assis, Thiago Santana Lemes, Guilherme Augusto Gomes De Villa.

**Formal analysis:** Marcus Fraga Vieira.

**Investigation:** Alfredo de Oliveira Assis, Thiago Santana Lemes, Guilherme Augusto Gomes De Villa.

**Methodology:** Marcus Fraga Vieira, Fábio Barbosa Rodrigues, Alfredo de Oliveira Assis, Thiago Santana Lemes, Guilherme Augusto Gomes De Villa.

**Supervision:** Marcus Fraga Vieira.

**Validation:** Marcus Fraga Vieira, Fábio Barbosa Rodrigues, Eduardo de Mendonça Mesquita, Rafael Reimann Baptista, Adriano de Oliveira Andrade.

**Visualization:** Marcus Fraga Vieira, Fábio Barbosa Rodrigues, Eduardo de Mendonça Mesquita, Rafael Reimann Baptista, Adriano de Oliveira Andrade.

**Writing – original draft:** Marcus Fraga Vieira, Fábio Barbosa Rodrigues, Alfredo de Oliveira Assis, Rafael Reimann Baptista, Adriano de Oliveira Andrade, Paula Hentschel Lobo da Costa.

**Writing – review & editing:** Marcus Fraga Vieira, Fábio Barbosa Rodrigues, Eduardo de Mendonça Mesquita, Rafael Reimann Baptista, Adriano de Oliveira Andrade, Paula Hentschel Lobo da Costa.

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
