## [Decision Letter · Decision Letter 0]

17 Feb 2021

PONE-D-20-35028

Effects of additional load at different heights on gait initiation: a statistical parametric mapping of center of pressure and center of mass behavior

PLOS ONE

Dear Dr. Baptista,

Thank you for submitting your manuscript to PLOS ONE. After careful consideration, we feel that it has merit but does not fully meet PLOS ONE’s publication criteria as it currently stands. Therefore, we invite you to submit a revised version of the manuscript that addresses the points raised during the review process.

We look forward to receiving your revised manuscript.

Kind regards,

YunJu Lee

Academic Editor

PLOS ONE

Journal Requirements:

Could you therefore please the title page into the beginning of your manuscript file itself, listing all authors and affiliations.

3. We note that your study is closely related to the following publication, on which you are an author:

https://www.sciencedirect.com/science/article/abs/pii/S0021929017301185?via%3Dihub

Although you have  cited the above study in the discussion section of your manuscript, we feel that the scientific rationale of the current study and the contribution that it makes to the field should be better justified.  Therefore, please cite and discuss the above study in the introduction section of your manuscript, clarifying how the present work is related to the previously published paper.

Please note that our second publication criterion states that "If a submitted study replicates or is very similar to previous work, authors must provide a sound scientific rationale for the submitted work and clearly reference and discuss the existing literature. Submissions that replicate or are derivative of existing work will likely be rejected if authors do not provide adequate justification." http://www.plosone.org/static/publication.action#results.

Thank you for your attention to this request.

"The study was partially supported by the governmental agencies CAPES (Finance Code 001), and CNPq."

Reviewers' comments:

Reviewer's Responses to Questions

**Comments to the Author**

1. Is the manuscript technically sound, and do the data support the conclusions?

Reviewer #1: Yes

Reviewer #2: Partly

2. Has the statistical analysis been performed appropriately and rigorously? 

Reviewer #1: Yes

Reviewer #2: N/A

3. Have the authors made all data underlying the findings in their manuscript fully available?

Reviewer #1: Yes

Reviewer #2: Yes

4. Is the manuscript presented in an intelligible fashion and written in standard English?

Reviewer #1: Yes

Reviewer #2: Yes

5. Review Comments to the Author

Reviewer #1: Please reinforce your theorical background with

- doi: 10.5312/wjo.v8.i11.815

And so after reading it please add to your analysis :

The calculation of stance leg stiffness conforming to Yiou et al.

- please also take into account the datas published by Honeine et al

https://doi.org/10.3389/fnhum.2016.00445

-In the method justify why the step lenght is not fixed because it could modified the calculation of MOS and BOS (please also add it in your analysis see 10.5312/wjo.v8.i11.815

Take into account the role of Tibialis Anterior on both anteroposterior and mediolateral side

-please argue at the end of the discussion the limitation in line with the references proposed

Please proposed clinical applications for MD and physical therapist Stance leg stiffness

See for example : please cited and argue in regards of https://doi.org/10.1038/s41598-018-19151-1

Reviewer #2: The purpose of this study was to compare the effect of asymmetric load at different height on APA phase COP and COM. This study used load corresponding to 10% body mass on swing foot, stance foot, and bilaterally with the load on waist and shoulder heights. The authors hypothesized that COP and COM behavior would change in the experimental conditions. With the comparison of COP displacement, COM placement, and pairwise SPM analysis, the COP and COM behavioral change were revealed and supported by the result.

This paper provides practical insight regarding the issues regarding material handling, particularly in the elder population. Some major comments and minor comments are listed below.

Major comments:

Was there any participant engaged in weight training? Since adaptions to weight training postures may influence the result, the background of the participants should be mentioned.

Line 112, please provide more details about the standing position in terms of the width of the feet. Was the participant keep the same standing position in all conditions?

How do you determine the load, 10% body mass, for the present study? Is there any reference?

Which kind of situation were you trying to simulate in this study, walking while carrying the backpack, the shoulder pack, or the tool belt? Please state the size of the load? Was the mass of the load evenly distributed? How to fix the load to the participant’s body? Would the load move when the participant was walking?

Please provide the results of statistical analysis of COP and COM displacements.

Minor comments:

Methods – please provide formula or references for COP and COM displacement calculation.

Line 112 – please state the reason for using barefoot in this study.

Results - What was the range of APA duration of the participants? Is the walking speed influence APA duration?

Figure 2 – is the term “task” in the horizontal axial meaning APA?

Line 356 – as described in the method, I would think only the tool belt is the application of this study.

6. PLOS authors have the option to publish the peer review history of their article (what does this mean?). If published, this will include your full peer review and any attached files.

Reviewer #1: **Yes: **Delafontaine Arnaud

Reviewer #2: No

---

## [Author Response · Author response to Decision Letter 0]

24 Mar 2021

Dr. YunJu Lee

Academic Editor

PLOS ONE

22/03/2021

Dear Dr Lee,

Re: Manuscript reference No. PONE-D-20-35028

Please find attached a revised version of our manuscript “Effects of additional load at different heights on gait initiation: a statistical parametric mapping of center of pressure and center of mass behavior”, which we would like to resubmit for publication as a scientific article in PLOS ONE. 

Your comments and those of the reviewers were highly insightful and enabled us to greatly improve the quality of our manuscript. In the following pages are our point-by-point responses to each of the comments of the reviewers as well as your own comments.

Revisions in the text are shown in the Revised Manuscript with Track Changes. We hope that the revisions in the manuscript and our accompanying responses will be sufficient to make our manuscript suitable for publication in PLOS ONE.

We shall look forward to hearing from you at your earliest convenience.

Yours sincerely,

Marcus Fraga Vieira, PhD

Please address all correspondence to:

Rafael Reimann Baptista, PhD

Pontifical Catholic University Rio Grande do Sul, Porto Alegre, Brazil

E-mail address: rafael.baptista@pucrs.br

 

Responses to the reviewers' comments on a point-by-point basis

Submission: PONE-D-20-35028

General Comments

We thank the reviewers for their thoughtful and in-depth comments regarding our manuscript. Their suggestions and remarks have helped us to reflect on the manuscript and prepare a better version of it. We thank the reviewers for their commitment and effort. We have carefully considered every comment and made the required changes. All comments will be answered and complemented in the new manuscript and some of them will be discussed here. Below we respond to the remarks on a point-by-point basis. The reviewer’s comment is provided first, and then we provide our answer in italic style.

Responses to the comments of Reviewer #1

Reviewer #1: Please reinforce your theorical background with - doi: 10.5312/wjo.v8.i11.815

Answer: Thank you for your suggestion. We have rewritten the Introduction section considering the Reviewer suggestion.

And so after reading it please add to your analysis:

The calculation of stance leg stiffness conforming to Yiou et al.

Answer: This study originates a very large amount of data due to several conditions we have assessed. Thus, we have to limit our analysis to make the article more understandable and more attractive to the reader. It was a decision we made at this moment, but we intend to extend our analysis to the other phases of gait initiation and include other variables of interest. Even so, the manuscript is over twenty pages long, so we prefer not to include an additional variable.

At this moment, therefore, we decide to focus only on the anticipatory postural adjustments (APA) phase and use a statistical method that considers the entire time series, avoiding extracting discrete variables.

For this reason, regarding stance leg stiffness, this calculation cannot be done. Such a model can only be applied to the swing phase of gait initiation, the execution phase from toe-off to foot contact. Please, see Yiou, E., Artico, R., Teyssedre, C. A., Labaune, O., and Fourcade, P. (2016a). Anticipatory postural control of stability during gait initiation over obstacles of different height and distance made under reaction-time and self-initiated instructions. Front. Hum. Neurosci. 10:449. doi: 10.3389/fnhum.2016.00449, and 

-In the method justify why the step length is not fixed because it could modify the calculation of MOS and BOS (please also add it in your analysis see 10.5312/wjo.v8.i11.815

Answer: This study aimed to assess the changes and strategies that potentially adjust the APA phase during gait initiation perturbed by asymmetrical load distribution at different heights. To not impose any kind of constraint that could compromise the adoption of such changes/strategies in APA phase the participants were instructed to assume a comfortable and natural standing position and to perform the gait initiation comfortably with their preferred foot at a self-selected speed. If Delafontaine et al. 2018 had fixed step length, they could not have observed the changes in step length due to the use of an orthosis (See Delafontaine, A., Fourcade, P., Honeine, J.L. et al. Postural adaptations to unilateral knee joint hypomobility induced by orthosis wear during gait initiation. Sci Rep 8, 830 (2018). https://doi.org/10.1038/s41598-018-19151-1)

These were the reasons we did not control step length. We have added some text, as follows:

Page 7, Lines 146-148

“The length of the first step was not controlled in order not to impose restrictions on any change in the task execution strategy due to the different conditions.”

Concerning the MoS and BoS calculation, as we have discussed above, at this moment, we have focused on the APA phase assessment, where the excursion of the extrapolated center of mass is small. MoS and BoS assessment will be useful when we examine the next phases of the gait initiation, especially at the first step foot contact at the end of the execution phase when the center of mass and extrapolated center of mass rapidly move toward foot contact of the swing foot.

- please also take into account the data published by Honeine et al

https://doi.org/10.3389/fnhum.2016.00445

Take into account the role of Tibialis Anterior on both anteroposterior and mediolateral side

Answer: As these comments are related, we have responded to them together. Following the Reviewer suggestion, we have added some text as follows:

Page 4, Lines 70-84

“ In the sagittal plane, the AP COP displacement during APA phase is mediated by the stereotyped activities of ankle muscles: the inhibition of the soleus and gastrocnemius that are followed by activation of the tibialis anterior in both legs [6,8]. The anticipatory soleus inhibition and tibialis anterior activation are not observed in all young healthy adults [9] and this functional variability of APA behavior is probably influenced by initial trunk posture (backward or forward inclination), speed of the first step and initial tonic muscle soleus activity [9].

In the frontal plane, the ML COP displacement during GI are controlled by coordinated action of swing leg hip abductors [2]. While COP moves backward by ankle muscles synergism, swing limb hip abductors move the COP to the swing limb [2] during the APA phase. Further studies have reported a slight knee and hip flexion of the support limb during APA that acts to unload the support limb complementing the action of the swing limb hip abductors [10]. In addition, it was observed that ankle dorsiflexors contribute to ML COP displacement during APA: coordinated activation of hip abductors and tibialis anterior during APA has a role in the ML COP displacement towards the swing foot [10].”

-please argue at the end of the discussion the limitation in line with the references proposed

Please proposed clinical applications for MD and physical therapist Stance leg stiffness

See for example : please cited and argue in regards of https://doi.org/10.1038/s41598-018-19151-1

Answer: Following the reviewer’s suggestions, we have added some text as follows:

Page 17, Lines 392-397

“The present study had some limitations. We only used kinetic data to describe the COP and COM displacement. Thus, we cannot make further assumptions about additional mechanisms that might be involved in force/impulse generation. Consequently, future studies should consider the simultaneous analysis of full body kinematic and electromyographic data to investigate the influence of different vertical positions of asymmetrical/symmetrical loads on the APA phase of gait initiation.”

Page 17, Lines 401-403

“The results found here can be used for manipulating/inducing gait initiation strategies in a clinical context, especially regarding cases of pathological patients unable to maintain a symmetrical body weight distribution [30–32].”

Page 17-18, Lines 404-411

“Additionally, since we have demonstrated here that asymmetrical overload can influence mediolateral postural stability during GI, and since mediolateral instability is believed to cause sideways falls, which can produce severe injuries in older adults [33], future studies should investigate if elderly people can adjust their APA to uneven extra load distribution on the body to preserve ML stability during GI. Such studies could provide the identification of APA alterations under challenging postural adaptations in older adults, could produce a better understanding of the mechanisms responsible for recurrent falls in older adults during gait initiation (17), and could even be used in rehabilitation protocols.”

Responses to the comments of Reviewer #2

Reviewer #2: The purpose of this study was to compare the effect of asymmetric load at different height on APA phase COP and COM. This study used load corresponding to 10% body mass on swing foot, stance foot, and bilaterally with the load on waist and shoulder heights. The authors hypothesized that COP and COM behavior would change in the experimental conditions. With the comparison of COP displacement, COM placement, and pairwise SPM analysis, the COP and COM behavioral change were revealed and supported by the result.

This paper provides practical insight regarding the issues regarding material handling, particularly in the elder population. Some major comments and minor comments are listed below.

Major comments:

Was there any participant engaged in weight training? Since adaptions to weight training postures may influence the result, the background of the participants should be mentioned. 

Answer: No, the participants were not engaged in any mode of physical training. This was included in page 6 line 126-127, as follows:

“This study enrolled 68 college students (32 males, 36 females; age: 23.65 ± 3.21 years; weight: 69.98 ± 8.15 kg; height: 1.74 ± 0.08 m), not engaged in any mode of physical training.”

Line 112, please provide more details about the standing position in terms of the width of the feet. Was the participant keep the same standing position in all conditions?

Answer: We have added some text to the manuscript, as follows:

Page 7, Lines 141-152

“The participants were instructed to assume a comfortable and natural standing position, barefoot, upper limbs alongside the trunk, one foot on each force platform. The participants were tested barefoot to avoid any influence of footwear and to make our results comparable to that of previous studies (12, 17-19). They were asked to stand as still as possible. After a verbal command, the participants started to walk with their preferred foot at a self-selected speed to the end of the walkway, performing two complete gait cycles. The length of the first step was not controlled in order not to impose restrictions on any change in the task execution strategy due to the different conditions. Next, they were asked to reposition themselves after each trial following the same instructions, taking the midline between the force platforms as a reference, and starting with the same foot. They executed as many familiarization trials as needed. The data acquisition started 2 s prior to the verbal command. The participants rested for a period of 30 s between each trial.”

How do you determine the load, 10% body mass, for the present study? Is there any reference?

Answer: 10% has been shown to be sufficient to modify the COM location in a standing position, and it was used in previous studies. We have added some references, as follows:

Page 7, Lines 157-158

“When present, the additional load was set to 10% of body mass (12,20).”

Wu, G., and MacLeod, M. (2001). The control of body orientation and center of mass location under asymmetrical loading. Gait Posture 13, 95–101. doi: 10.1016/s0966-6362(00)00102-8

Caderby T, Yiou E, Peyrot N, de Viviés X, Bonazzi B, Dalleau G. Effects of changing body weight distribution on mediolateral stability control during gait initiation. Front Hum Neurosci. 2017;11(127).

Which kind of situation were you trying to simulate in this study, walking while carrying the backpack, the shoulder pack, or the tool belt?

Answer: Thank you for your comment. We are not trying to simulate a specific situation. We are trying to understand the strategies to maintain a safe first step during gait initiation and how the individual dynamically adapts these strategies to face different situations.

In previous studies, we have investigated, for example, the adaptations during gait initiation when the individual is on a sloped surface (see Vieira MF, de Brito AA, Lehnen GC, Rodrigues FB. Center of pressure and center of mass behavior during gait initiation on inclined surfaces: A statistical parametric mapping analysis. J Biomech. 2017;56(3):10–8).

In the present study, we are interested to verify the strategies adopted by the individual when the distribution of body weight (center of mass) between both legs is changed. Also, we are interested in analyzing how a change in the height of the center of mass influences gait initiation, and what are the strategies adopted to a safe first step.

In this sense, we focused on analysing the anticipatory postural adjustments, which are considered a feedforward mechanism responsible for controlling mediolateral stability during gait initiation.

Of course, the situations in both studies are common daily situations faced by the individuals. Sloped surfaces represent a common postural challenge in human daily activities and represent a potential risk of fall for some populations. By contrast, concerning the distribution of body weight, as Caderby et al. 2017 said, “In able-bodied subjects, natural posture with asymmetrical body weight distribution between the legs can be found in ecological situations, typically when one side of the body is loaded with an additional mass (e.g., carrying an object with a single hand or a backpack on one shoulder, etc.)”.

The results/conclusions can be used to manipulate/induce gait initiation strategies in a clinical context.

Please state the size of the load? Was the mass of the load evenly distributed? How to fix the load to the participant’s body? Would the load move when the participant was walking?

Answer: The protocol was similar to that reported in Caderby et al. 2017. The load was fixed to the individual's body using a belt in a well-adjusted way, in which weights were attached to reach the desired load, small bags contained sand evenly distributed. The load was made as small as possible.

We have added some text to the manuscript as follows:

Page 7, Lines 157-159

“When present, the additional load was set to 10% of body mass (12,20). The load consisted of a belt firmly positioned at the waist (close to the COM’s position) or at the shoulder. Weights were attached to the belt to carefully reach the desired load.”

Please provide the results of statistical analysis of COP and COM displacements. 

Answer: The results are shown in Figures 4-9. It should be noted that different from usual methods, in which discrete features of the signal are extracted, we have used a statistical analysis called statistical parametric mapping, using the entire time series corresponding to the APA phase of the GI. These statistics are explained in lines 186-190, pages 8-9 and the way to interpret the figures is described in lines 203-205, page 9. The statistical results when using statistical parametric mapping are presented graphically.

Minor comments:

Methods – please provide formula or references for COP and COM displacement calculation.

Answer: We did not provide the formula for COP calculation because they are widely reported in the literature. The COM calculation was explained in lines 180-183. Most important, several systems provide the outcomes for COP both in each force platform and the resultant COP between the force platforms. This is our case. In the Vicon system, the cameras and the force platforms are operated synchronously and the Vicon software provides the outcomes for COP. There is no need to export force platform data to calculate the COP. The force platform data was exported only for COM calculation. We have added references and some text to the manuscript as follows:

Page 6, Lines 137-140

“Both kinetic and kinematic data were captured by a motion capture system (Vicon Motion Systems Ltd, Oxford, UK) comprising 10 infrared cameras and the two force platforms, operating synchronously at 100 Hz.”

Page 6, Lines 173-183

“The motion capture system software (Vicon Nexus, version 2.11, Vicon Motion Systems Ltd, Oxford, UK) provided the COP, ground reaction forces (GRF), and kinematic time series. The raw force platform and kinematic data were filtered using a fourth-order, zero-lag, low-pass Butterworth filter with a cutoff frequency of 12.5 Hz (14). The COP, GRF, and kinematics time series were exported in txt file (a full description of how to calculate the COP can be found elsewhere (8,15)). Next, the COP time series were normalized such that at the beginning of the task, the AP and ML COP components were described in relation to the mid-point between the lateral malleoli (i.e., in relation to a foot coordinate system). The projection of COM on the force platform was estimated using a double integration of the previously exported GRFs by the trapezoidal method (14) using a custom-written Matlab code (© 1994-2021 The MathWorks, Inc.).

Line 112 – please state the reason for using barefoot in this study.

Answer: In a previous study, we have shown the influence of shoes on gait initiation (please, see Vieira MF, Sacco IdCN, Nora FGdSA, Rosenbaum D, Lobo da Costa PH (2015) Footwear and Foam Surface Alter Gait Initiation of Typical Subjects. PLoS ONE 10(8): e0135821. doi:10.1371/ journal.pone.0135821). Besides, the participants often wear different types of footwear. Thus, to avoid the influence of footwear on gait initiation and to make our study comparable to other studies (see Caderby et al. 2017, Khanmohammadi et al. 2017, for example), we have tested the participants barefoot. We have added some text to the manuscript, as follows:

Page 7. Lines 142-144

“The participants were tested barefoot to avoid any influence of footwear and to make our results comparable to that of previous studies (12,17–19).”

Results - What was the range of APA duration of the participants?

Answer: As we have explained above to Reviewer #1, this study originates a very large amount of data. At this moment, we have prioritized analysing the entire APA phase time series, and not extracting discrete variables. Thus, as we intended to make an SPM analysis of the entire time series, we did not present this information, since SPM requires the data to be time-normalized. Thus, COP and COM displacements during APA phase were time-normalized to 0-30% of the task. The APA duration ranged 0.2721-0.3510 s.

Is the walking speed influence APA duration?

Answer: Yes, gait speed influences the APA. However, as APA is related to the safety and stability of the first step, the participants were tested initiating gait at a self-selected, comfortable, and natural speed in all conditions. The self-selected walking speed, from a biomechanical point of view, provides the most stable walking.

Figure 2 – is the term “task” in the horizontal axial meaning APA? 

Answer: No, it is the task (gait initiation). Observe that the figure shows a 0 to 30% segment of the task, that approximately corresponds to APA. This was explained in lines 191-192, page 8.

Line 356 – as described in the method, I would think only the tool belt is the application of this study. Answer: In the methods section, we described the overload positioned symmetrically and asymmetrically at different heights, and we have emphasized this in the title of the manuscript. However, as we have discussed above, we are not trying to simulate a specific situation. We are trying to understand the strategies to maintain a safe first step during gait initiation and how the individual dynamically adapts these strategies to face different situations: asymmetrical/symmetrical overload at different heights. Of course, the situations can be found in daily situations faced by the individuals. As Caderby et al. 2017 said, “In able-bodied subjects, natural posture with asymmetrical body weight distribution between the legs can be found in ecological situations, typically when one side of the body is loaded with an additional mass (e.g., carrying an object with a single hand or a backpack on one shoulder, etc.)”. Additionally, the results/conclusions can be used to manipulate/induce gait initiation strategies in a clinical context, regarding cases of pathological patients (see, for example, Marigold, D. S., and Eng, J. J. (2006). The relationship of asymmetric weight bearing with postural sway and visual reliance in stroke. Gait Posture 23, 249–255. doi: 10.1016/j.gaitpost.2005.03.001, or Tessem, S., Hagstrøm, N., and Fallang, B. (2007). Weight distribution in standing and sitting positions and weight transfer during reaching tasks, in seated stroke subjects and healthy subjects. Physiother. Res. Int. 12, 82–94. doi: 10.1002/pri.362).

---

## [Decision Letter · Decision Letter 1]

24 May 2021

Effects of additional load at different heights on gait initiation: a statistical parametric mapping of center of pressure and center of mass behavior

PONE-D-20-35028R1

Dear Dr. Baptista,

We’re pleased to inform you that your manuscript has been judged scientifically suitable for publication and will be formally accepted for publication once it meets all outstanding technical requirements.

Kind regards,

YunJu Lee

Academic Editor

PLOS ONE

Additional Editor Comments (optional):

Reviewers' comments:

Reviewer's Responses to Questions

**Comments to the Author**

1. If the authors have adequately addressed your comments raised in a previous round of review and you feel that this manuscript is now acceptable for publication, you may indicate that here to bypass the “Comments to the Author” section, enter your conflict of interest statement in the “Confidential to Editor” section, and submit your "Accept" recommendation.

Reviewer #2: All comments have been addressed

2. Is the manuscript technically sound, and do the data support the conclusions?

Reviewer #2: Yes

3. Has the statistical analysis been performed appropriately and rigorously? 

Reviewer #2: Yes

4. Have the authors made all data underlying the findings in their manuscript fully available?

Reviewer #2: Yes

5. Is the manuscript presented in an intelligible fashion and written in standard English?

Reviewer #2: Yes

6. Review Comments to the Author

Reviewer #2: (No Response)

7. PLOS authors have the option to publish the peer review history of their article (what does this mean?). If published, this will include your full peer review and any attached files.

Reviewer #2: **Yes: **Sheng-Chieh Yang

---

## [Editor Report · Acceptance letter]

2 Jun 2021

PONE-D-20-35028R1 

Effects of additional load at different heights on gait initiation: a statistical parametric mapping of center of pressure and center of mass behavior 

Dear Dr. Baptista:

I'm pleased to inform you that your manuscript has been deemed suitable for publication in PLOS ONE. Congratulations! Your manuscript is now with our production department. 

Kind regards, 

on behalf of

Dr. YunJu Lee 

Academic Editor

PLOS ONE